# Erector Spinae Plane Blocks for the Early Analgesia of Rib Fractures in Trauma (ESPEAR): protocol for a multicentre pilot randomised controlled trial with feasibility and embedded qualitative assessment

David W Hewson [1,2] Jessica Nightingale,[3] Reuben Ogollah,[4] Benjamin J Ollivere,[2,3] Matthew L Costa,[5] Simon Craxford [2] Peter Bates,[6] Nigel M Bedforth[1]

For numbered affiliations see end of article.

**Correspondence to**
Dr David W Hewson;
david.hewson@nottingham.ac.uk

## ABSTRACT

**Introduction** Patients with rib fractures commonly experience significant acute pain and are at risk of hypoxia, retained secretions, respiratory failure and death. Effective analgesia improves these outcomes. There is widespread variation in analgesic treatments given to patients including oral, intravenous and epidural routes of administration. Erector spinae plane (ESP) blockade, a novel regional analgesic technique, may be effective, but high-quality evidence is lacking.

**Methods and analysis** To determine if a definitive trial of ESP blockade in rib fractures is possible, we are conducting a multicentre, randomised controlled pilot study with feasibility and qualitative assessment. Fifty adult patients with rib fractures will be randomised in a 1:1 ratio to ESP blockade with multimodal analgesia or placebo ESP blockade with multimodal analgesia. Participants and outcome assessors will be blinded. The primary feasibility outcomes are recruitment rate, retention rate and trial acceptability assessed by interview.

**Ethics and dissemination** The study was approved by the Oxford B Research Ethics Committee on 22 February 2022 (REC reference: 22/SC/0005). All participants will provide written consent. Trial results will be reported via peer review and to grant funders.

**Trial registration number** ISRCTN49307616.

## STRENGTHS AND LIMITATIONS OF THIS STUDY

⇒ There is widespread variation in the care of patients following rib fractures. The clinical effectiveness of erector spinae plane (ESP) blocks and catheters in this patient group is unclear.

⇒ This is a feasibility study with piloting of candidate clinical outcome measures to determine if a definitive trial is feasible; the present work alone cannot answer whether ESP blocks are an effective analgesic modality for patients with rib fractures.

⇒ The study will test if an analgesic placebo arm is an acceptable methodological feature for participants, clinicians and investigators.

⇒ The study uses a programmed intermittent bolus regime for local anaesthetic delivery, but the ideal dose and method of delivery for ESP blocks remains unknown. The study will not answer this question.

⇒ The study used a double-blind design (patients and outcome assessors) and the effectiveness of blinding will be determined by patient and staff interviews.

## INTRODUCTION

The pain from rib fractures is often described by patients as the worst pain they have ever experienced. The major complication of this pain is that patients are unable to cough and breathe deeply, causing atelectasis, retained secretions, hypoxaemia, pneumonia and progressive respiratory failure. Deterioration may require mechanical ventilation on an intensive care unit (ICU) and lead to death.[1 2] This morbidity and mortality are a direct result of severe pain and impaired gas exchange from underlying contused lung parenchyma and altered ventilatory mechanics from the bone injury.[3] The presence of rib fractures in trauma is associated with a significantly increased risk of death, regardless of other injuries, with ORs of 1.4 (95% CI 1.3 to 1.6) for adults 18–45 years old and 2.5 (95% CI 2.3 to 2.8) for adults older than 64 years.[4] This injury is therefore particularly devastating for older adults who have a higher risk of death and are likely to sustain rib fractures from less traumatic accidents (due to bone fragility), for example, falling from standing height.[5]

A key objective in the multidisciplinary care of people with rib fractures is the assessment and treatment of pain to provide patient comfort and allow normal respiration and cough to minimise the risk of respiratory failure.[3 6] Alongside specialist physiotherapy and daily multidisciplinary review, good pain management is a vital element of early rib fracture care. Despite this, there is no agreement about the optimal pain relief to give patients. The literature on the use of the different analgesic techniques in rib fractures is inconclusive. Although national and international guidance recommends a multimodal approach in preference to opioid medications alone,[7] two meta-analyses concluded that the evidence to recommend any specific treatment modality is insufficient, and that there is no firm evidence for benefit or harm of one analgesic technique over another.[8 9] This leaves clinicians unsure of which analgesic techniques to use. National UK guidance specifies protocolised analgesic regimes as a standard of care for every patient with multiple rib fractures.[10] However, the paucity of evidence meant that this guidance could not recommend which analgesic modality (epidural, peripheral nerve blocks or opioid) should be used in which clinical circumstances. Most patients with rib fractures are given a combination of analgesic drugs like paracetamol, non-steroidal anti-inflammatories, opioids and ketamine to help them cope with severe pain; these are the cornerstones of multimodal analgesia in this setting. Medication side effects (including nausea, pruritus, hallucinations, constipation, renal failure and respiratory depression) significantly limit their use. Some patients receive thoracic epidural analgesia (TEA) and some receive other forms of regional anaesthesia nerve blocks, but the delivery of these interventions by pain specialist anaesthetists is driven more by local expertise and experience than by high-quality evidence.[11]

Regional anaesthesia (including nerve blocks) are clinically useful following rib fractures due to their opioid-sparing effect (therefore reducing serious drug-related side effects) and the superior dynamic pain relief they provide. Traditional techniques to block the thoracic nerve supply to the ribs include TEA, paravertebral blockade and intercostal blockade. Systematic review and meta-analysis of these techniques suggested that TEA provides good pain relief; however, this benefit does not translate into superior outcomes such as occurrence of pulmonary complications and length of time spent in hospital, intensive care or requiring mechanical ventilation.[12] Unfortunately, TEA has a significant failure rate and is also associated with common and potentially catastrophic complications[13] leading to permanent paralysis, and is therefore contraindicated in approximately one-fifth of people with significant injuries. TEA is a complex intervention to perform, practised by a small and reducing number of anaesthetists nationally and is not available equitably to patients. Even within a single hospital the care delivered varies depending on the time of day and availability of staff to perform such a complex

analgesic technique. Only an estimated 9.9%–18.4% of patients receive TEA for rib fracture pain.[14]

The erector spinae plane (ESP) block is a regional anaesthetic technique involving the infiltration and infusion of local anaesthetic along fascial planes containing dorsal and ventral rami of thoracic spinal nerves supplying the chest wall.[15] The injection is performed away from the spinal cord (thereby avoiding the complications of TEA). ESP blocks were first described in 2016[16] and have demonstrated analgesic efficacy for patients on enhanced recovery after surgery protocols following spinal,[17] breast,[18] thoracic[19] and cardiac surgeries.[20] In these postoperative acute pain settings, ESP blocks have been shown to reduce patient-reported pain scores and opioid consumption significantly in the early postoperative period compared with multimodal analgesia regimes alone. However, the role of ESP blocks in the management of acute rib fracture pain is currently uncertain.[21] There are no experimental pragmatic multicentre trials published in this setting; however, single-centre cohort data demonstrate ESP blocks provide effective pain relief and improve respiratory function when added to multimodal analgesia in patients with rib fractures.[15 22] Higher quality clinical evidence is urgently needed to guide clinicians on whether the ESP block is a suitable addition to current multimodal analgesia in patients with rib fractures. A definitive trial on this topic would promote evidence-based practice in rib fracture management and reduce unnecessary variation in clinical practice across UK trauma centres. However, there is currently not enough evidence on the effectiveness and acceptability of ESP blocks for rib fractures to undertake a definitive randomised controlled trial (RCT).

The aim of this study is therefore to determine if it is feasible to undertake a definitive RCT to establish if ESP blocks are a clinically effective early treatment for acute pain in patients hospitalised with rib fractures. Formal hypothesis testing for effectiveness or efficacy is not undertaken in feasibility studies. The aim of this trial is not to assess effectiveness or efficacy but to determine feasibility of progression to a definitive RCT.

## METHODS AND ANALYSIS
### Objectives
This study has the primary objective of determining whether it is feasible to undertake a definitive RCT to establish if ESP blocks are a clinically effective early treatment for acute pain in patients hospitalised with rib fractures. Our primary objectives are to determine:

▶ Trial recruitment rate.
▶ Trial retention rate.
▶ Barriers and facilitators to recruitment and retention among participants and recruitment site staff (anaesthetists, allied health professionals, surgeons and research staff) with regard to the acceptability of the trial intervention.

Secondary trial objectives are:

- ► To determine the willingness of anaesthetists to randomise patients to intervention or control and willingness of potential participants to randomisation.
- ► To identify causes of protocol violation and trial withdrawal.
- ► To assess the completeness of data arising from the trial.
- ► To assess the fidelity of the trial intervention in terms of ESP catheter dislodgement, blockage or other technical failure.
- ► To assess the acceptability of the intervention to participants.
- ► To describe complications of the intervention.
- ► To pilot the collection of candidate outcome measures for a future definitive trial.
- ► To determine the preliminary indicators of effectiveness as measured by candidate clinical outcome measures.

## Patient and public involvement

The study question builds on previous qualitative work undertaken to validate a patient-derived recovery scale. The scale was developed following interviews with 50 patients and health professionals, with subsequent validation in a 250-patient study. This work characterised the experience of pain and breathing difficulties following rib fracture; identifying management of pain as a research priority for this patient population. The outcome scale developed through this study will be used as an outcome measure in this trial, to help capture patient-centred outcomes. Specifically for this study we have facilitated virtual focus groups with patients who have previously sustained rib fractures and were admitted to Nottingham University Hospitals NHS Trust. The groups discussed the following aspects: the question and study design; recruitment and consent; follow-up data collection; acceptability of blinding; and preferred outcome measures. There was strong support for this study; with individuals acknowledging that pain management of non-operatively managed injuries was an important but often overlooked area of their care. A new regional anaesthetic technique was perceived to be valuable as a treatment option or adjunct since participants said they would be keen to avoid the side effects associated with oral and intravenous analgesia. The ESP block was perceived by focus group members as less invasive than an epidural. The inclusion of a sham intervention was discussed and deemed acceptable given the integrity of the research. The proposed outcome measures were reviewed by participants and were felt to be comprehensive. They valued the addition of embedded qualitative work within the study to allow for holistic feedback about study acceptability for patients and staff.

## Population and setting

The target population is patients newly admitted to the Major Trauma Centre (MTC) with one or more new rib fractures who can receive the trial intervention within 12 hours of admission to hospital. Participants will be recruited via their usual clinical care teams (emergency department, major trauma and/or acute pain services), who will notify study investigators of a potentially eligible participant for screening and recruitment purposes.

### Inclusion criteria

- ► Age ≥18.
- ► New admission to major trauma centre and can receive trial intervention within 12 hours of admission.
- ► Mechanism of injury blunt thoracic trauma.
- ► Radiographic evidence of one or more new traumatic rib fractures.
- ► Moderate or severe unilateral acute pain (defined as 11-point numerical rating scale (NRS) pain >4 when patient is performing vital capacity breath or effective cough) at time of enrolment. Patients may have bilateral fractures, but pain must be unilateral.

### Exclusion criteria

- ► Patient refusal or inability to give informed written consent for any reason.
- ► Thoracic injury requiring emergent operative or interventional radiology management.
- ► Allergy to local anaesthetic.
- ► Infection at site of ESP block.
- ► Actual or estimated total body weight ≤50 kg thereby precluding safe dosing of local anaesthetic for ESP block.

## Interventions and blinding

Following written consent, participant randomisation will be performed to a 1:1 ratio using a web-based automated computer-generated minimisation algorithm with treatment groups balanced for: age, gender, polytrauma and unilateral or bilateral rib fractures. Other than the allocated intervention, both groups will be followed up in the same way to exclude bias beyond procedures necessary for the allocation treatment. Randomisation will be to two groups:
1. ESP block and catheter plus multimodal analgesia (intervention).
2. Sham ESP block and catheter plus multimodal analgesia (control).

### ESP block plus multimodal analgesia

Participants randomised to ESP block plus multimodal analgesia (intervention) will receive a US-guided ESP block and catheter targeting the vertebral transverse process corresponding to the midpoint of the consecutively fractured ribs on the side of pain. An initial fascial plane injection of 30 mL of 0.25% levobupivacaine will be placed, followed by catheter-delivered programmed intermittent boluses of 15 mL 0.125% levobupivacaine given every 3 hours with optional patient or clinician bolus of 5 mL every 1 hour.

Participants allocated to intervention will additionally receive standard supportive care and multimodal analgesia according to British Orthopaedic Association 2016

guidelines. The site-specific adoption of multimodal analgesia regimes will be reviewed as part of the site feasibility.

### Sham ESP block plus multimodal analgesia

Participants randomised to sham ESP block plus multimodal analgesia (control) will receive a sham/placebo ultrasound-guided ESP block and catheter targeting the vertebral transverse process corresponding to the midpoint of the consecutively fractured ribs on the side of pain. A single 1 mL subcutaneous injection of saline 0.9% will be made and a perineural catheter applied and affixed by skin glue externally on the skin which will be dressed and connected to an infusion pump with patient button which will remain turned off. Participants allocated to control will additionally receive standard supportive care and multimodal analgesia according to individual trial site protocol as per the intervention arm.

Participants in both arms will continue to receive multimodal analgesia as dictated by their usual clinical care team. Following Erector Spinae Plane Blocks for the Early Analgesia of Rib Fractures in Trauma enrolment, additional regional anaesthetic techniques (eg, thoracic epidural insertion) will be undertaken at the discretion of the treating clinician, will be recorded in the trial Case Report Form (CRF) and will not lead to participant withdrawal.

### Blinding

Participants will be blinded to group allocation. Placebo effects are known to play a significant role on pain perception and patient expectation of analgesic efficacy; therefore, it is important that a definitive trial includes a placebo arm. This pilot RCT will test this blinding effectiveness as part of the feasibility-embedded qualitative process analysis. Anaesthetists siting the ESP block or sham ESP block will not be blinded to group allocation, since this is not technically possible. Outcome assessors will be blinded to group allocation. Blinding will be achieved by the infusion devices in both arms being stored in a black carry case during the infusion.

### Outcome measures

#### Primary feasibility outcomes

The primary feasibility outcomes which will be measured to meet the objectives of this trial are:

► Recruitment rate. Defined as the number of eligible participants who consent to participate in the trial as a percentage of all eligible participants. This will be presented per centre per month and measured over the recruitment period (from randomisation of the first participant to randomisation of the final participant). The target recruitment rate is defined as recruitment of 50 participants from three recruiting centres, with each centre being open to recruitment for 12 months. This produces a mean trial target recruitment rate of 1.4 participants per centre per month.

► Retention rate. Defined as the proportion of randomised participants who complete a 6-week follow-up with valid candidate clinical outcome data (see below).

► Barriers and facilitators to recruitment and retention among participants and recruitment site staff (anaesthetists, allied health professionals, surgeons and research staff). This will be assessed in the embedded qualitative study.

#### Secondary feasibility outcomes

The secondary feasibility outcomes are as follows:

► Trial eligibility rate. Defined as the proportion of those patients screened who were eligible for enrolment in the trial.

► Trial consent rate. Defined as the proportion of eligible patients who provided written consent for inclusion in the trial.

► Willingness of anaesthetists to randomise patients to intervention or control and willingness of potential participants to randomisation. This will be achieved through qualitative evaluation, including scrutiny of screening logs, completion of an open-ended survey with healthcare staff and qualitative interviews with research staff conducted by the central research team.

► Causes of protocol violation. Causes will be identified from the Investigator Site File.

#### Secondary clinical outcomes

The following clinical outcomes are considered secondary outcomes of the trial, and will be measured to assess the relevance, completeness and acceptability of these outcomes for use in a future definitive RCT:

► Static chest wall pain intensity. Measured on Short-Form McGill Pain Questionnaire 2 to describe the worst pain experienced by the patient between the following eight time points: 24 hours prior to receipt of the trial intervention (defined as trial baseline), then at 24, 48 and 72 hours. Scores will be described at each time interval in comparison to baseline and summed to provide a cumulative static chest wall pain score. The time at which each measure is taken will also be recorded.

► Functional (ie, dynamic) chest wall pain intensity. Measured on a modified Functional Pain Scale (m-FPS) as the worst pain experienced by the patient during the following eight time points: 24 hours prior to receipt of the trial intervention (defined as trial baseline), then at 24, 48 and 72 hours. The time at which each measure is taken will also be recorded. Scores will be described at each time interval in comparison to baseline and summed to provide a cumulative functional chest wall pain score. The m-FPS consists of the following Likert-scaled responses: 0=no pain; 1=tolerable pain but able to perform vital capacity breath and effective cough; 2=tolerable pain but prevents either vital capacity breath or effective cough; 3=intolerable pain but can perform either vital capacity breath

or effective cough; 4=intolerable pain and unable to perform vital capacity breath or effective cough; 5=intolerable and unable to verbally communicate due to pain.

► Forced vital capacity, forced expiratory volume in 1 s and peak cough flow (spirometry). Measured by bedside portable spirometry. Measured immediately prior to receipt of trial intervention (defined as trial baseline), then at the following time points following receipt of intervention: 3, 6, 9, 12, 24, 48 and 72 hours.

► Cumulative non-opioid analgesic consumption. The administration of the non-opioid analgesics paracetamol and non-steroidal anti-inflammatories will be measured as total doses administered in the 24 hours prior to receipt of the trial intervention (defined as trial baseline), then at the following time points following receipt of the intervention: 24, 48 and 72 hours.

► Cumulative opioid analgesic consumption. The administration of the opioid analgesics will be measured as total dose administered in the 24 hours prior to receipt of the trial intervention (defined as trial baseline), then at the following time points following receipt of the intervention: 24, 48 and 72 hours. All doses will be converted to morphine equivalents for analysis.

► Cumulative ketamine analgesic consumption. The administration of the ketamine will be measured as total dose administered in the 24 hours prior to receipt of the trial intervention (defined as trial baseline), then at the following time points following receipt of the intervention: 24, 48 and 72 hours.

► Additional procedures of regional anaesthesia following ESP block. The administration of the following additional procedures of regional anaesthesia will be recorded in the 24 hours prior to receipt of the trial intervention (defined as trial baseline), then at the following time points following receipt of the intervention: 24, 48 and 72 hours: intercostal, pleural, serratus plane, non-trial erector spinae, paravertebral and epidural blockade.

► Opioid-related side effects. The following opioid-related side effects will be assessed immediately prior to receipt of trial intervention (defined as trial baseline) then at the following time points following receipt of intervention; 24, 48 and 72 hours:
Constipation, defined as absence of bowel movement in the preceding 24-hour period.
Nausea or vomiting, scored on a 5-point scale (0=no nausea or vomiting; 1=mild nausea, no treatment required; 2=nausea, antiemetics administered; 3=vomiting, antiemetics administered; 4=nausea or vomiting unresponsive to antiemetic therapy).
Pruritus, scored on 11-point NRS.
Opioid-induced sedation, scored on Modified Observer's Assessment of Alertness/Sedation Scale.

► Oxygen requirement. Measured as maximum flow rate of supplemental oxygen administered to participant

immediately prior to receipt of trial intervention (defined as trial baseline), then at the following time points following receipt of intervention; 3, 6, 9, 12, 24, 48 and 72 hours.

► Complications of regional anaesthesia. The following complications of regional anaesthesia will be assessed at 24, 48 and 72 hours following receipt of intervention:
Treatment for local anaesthetic toxicity, defined as administration of intralipid therapy in the preceding 24-hour period.
Bleeding or infection at intervention insertion site.
Catheter dislodgement requiring resited intervention in preceding 24-hour period.

► Condition-specific outcome measure. Measured on Outcomes after Chest Trauma Score (OCTS) to describe severity of rib-related symptoms (domains include mobility, breathing, activities, personal care, well-being and pain). The OCTS will be administered twice prior to receipt of trial intervention (defined as trial baseline) then at the following time points following receipt of intervention: 72 hours and 6 weeks.

► Diagnosis of pneumonia. Defined as administration of antibiotics for community-acquired or hospital-acquired pneumonia assessed in the 24 hours prior to receipt of trial intervention (defined as trial baseline) then at the following time points following receipt of intervention; 24, 48 and 72 hours and 6 weeks.

► Escalation of care to critical care. Defined as admission to level 2 (High Dependency Unit (HDU)) or level 3 (Intensive Care Unit (ICU)) bed assessed in the 24 hours prior to receipt of trial intervention (defined as trial baseline) then at the following time points following receipt of intervention; 24, 48 and 72 hours and 6 weeks.

► Length of hospital stay. Assessed 6 weeks following receipt of intervention.

► Quality of life measured on 5-Level version of Euro-Qol-5 Dimension. Assessed in the 24 hours prior to receipt of trial intervention (defined as trial baseline), then at 72 hours and 6 weeks following receipt of intervention.

► All-cause mortality. Assessed 6 weeks following receipt of intervention.

## Sample size calculation

Formal sample size calculation is not appropriate for feasibility studies. Currently, there is no single-agreed method for sample size for a feasibility trial, but most authors propose a sample size between 24 and 60 depending on the study aims.[23 24] To answer our key objectives, we aim to recruit 50 participants, allowing estimation of recruitment and retention rates with a margin of error of less than 10%.

## Statistical analysis

Data will be collected via REDCap database. Data analysis will primarily be descriptive to address the feasibility objectives of the trial. All analyses will be documented in a statistical analysis plan which will be finalised prior to database lock. Feasibility outcomes will be estimated using descriptive statistics (with 95% CIs) and will include screening rates, recruitment rates, follow-up rates, protocol adherence and amount of missing data for clinical outcomes. Key baseline characteristics (age, sex) will be compared between trial participants and the ineligible and non-consenting patients, to ascertain adequacy of inclusion/exclusion criteria and likely generalisability of the trial to the required targeted population. Similarly, we will compare the key patient characteristics between those followed up and those lost to follow-up and investigate how similar this is across the treatment arms to assess possible attrition bias in data collection. A baseline table will compare important demographic and clinical characteristics between the two treatment arms. It is not a primary objective of the feasibility trial to obtain definitive estimates of intervention effect on clinical outcomes and so the clinical outcomes will be analysed descriptively. Additionally, we will use appropriate regression method to estimate the likely range of intervention effects (point estimate and CIs) for key clinical outcomes adjusted for minimisation variables. Reporting of the study will be according to the Consolidated Standards of Reporting Trials statement: 2016 extension to randomised pilot and feasibility trials.[25]

## ETHICS AND DISSEMINATION

The study was granted approval by the Oxford B Research Ethics Committee on 22 February 2022 (REC reference: 22/SC/0005). Substantial amendments that require review by REC will not be implemented until the REC grants a favourable opinion for the trial. All personal identifiable information collected during the trial will be coded, depersonalised with unique codes for each patient. The trial will be compliant with the requirements of the General Data Protection Regulation 2018 and the Data Protection Act 2018. The chief investigator and principal investigators at participating sites will have access to the full data set. Relevant anonymised patient-level data will be made available on reasonable request. Day-to-day trial management will be provided by the Trial Management Group, who will meet at least once per month. Independent oversight of trial conduct will be provided by a Trial Steering Committee, attended by the trial chief investigators and methodologist, with three independent members with expertise in trial methodology and statistics, anaesthesia and trauma care.

A manuscript for a high-impact peer-reviewed journal will be prepared. Authorship will be determined in accordance with International Committee of Medical Journal Editors guidelines,[26] and other contributors will be acknowledged. The results of this project will be disseminated to patients through local mechanisms at all participating centres.

**Author affiliations**
¹Department of Anaesthesia, Nottingham University Hospitals NHS Trust, Nottingham, UK
²Academic Unit of Injury, Recovery and Inflammation Sciences, School of Medicine, University of Nottingham, Nottingham, UK
³Trauma and Orthopaedics, Nottingham University Hospitals NHS Trust, Nottingham, UK
⁴Nottingham Clinical Trials Unit, University of Nottingham, Nottingham, UK
⁵Nuffield Department of Orthopaedics, Rheumatology and Musculoskeletal Sciences, Oxford University, Oxford, UK
⁶Barts Health NHS Trust, London, UK

**Contributors** DWH and NMB conceived the study, refined the study design, are responsible for ongoing trial management and wrote and approved this protocol. JN, RO and MLC refined the study design, are responsible for ongoing trial management and wrote and approved this protocol. BJO conceived the study, is responsible for ongoing trial management and wrote and approved this protocol. SC and PB are responsible for ongoing trial management and wrote and approved this protocol.

**Funding** This work was funded by the National Institute for Health Research (NIHR) (Research for Patient Benefit Programme Call 42 (project reference: NIHR202195)).

**Disclaimer** The views expressed are those of the authors and not necessarily those of the NIHR or the UK Department of Health and Social Care.

**Competing interests** MLC is an NIHR senior investigator.

**Patient and public involvement** Patients and/or the public were involved in the design, or conduct, or reporting, or dissemination plans of this research. Refer to the Methods section for further details.

**Patient consent for publication** Not applicable.

**Provenance and peer review** Not commissioned; externally peer reviewed.

**ORCID iDs**
David W Hewson http://orcid.org/0000-0002-5314-8522
Simon Craxford http://orcid.org/0000-0002-4672-4587

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
