## [Reviewer comments · BMJ Open]

ARTICLE DETAILS

TITLE (PROVISIONAL)	Erector Spinae Plane blocks for the Early Analgesia of Rib fractures in trauma (ESPEAR): protocol for a multicentre pilot randomised controlled trial with feasibility and embedded qualitative assessment
AUTHORS	Hewson, David; Nightingale, Jessica; Ogollah, Reuben; Ollivere, Benjamin; Costa, Matthew; Craxford, Simon; Bates, Peter; Bedforth, Nigel

VERSION 1 – REVIEW

REVIEWER	Carver, Thomas Medical College of Wisconsin, Surgery
REVIEW RETURNED	16-Apr-2022

GENERAL COMMENTS	I feel that the authors present a very clear description of their study and that it is an important topic that will add to the gap in literature. Recognizing that the study protocol is already in place I still have some suggestions/questions regarding the design that I feel should be reviewed. Page 10 line 59. I do not know what a “guided ESP block story” is. Please clarify this sentence. Study design. 1 rib fracture may be too inclusive. Additionally, what about location of fractures? Are you going to try to do a ESP on someone with a first rib fracture? Do you think the anesthetic will dissect up to that level? I am worried that this will severely confound your study and I would suggest that 3 or more rib fractures be considered instead. You will be more likely to have a difference in clinical outcomes and this will also better inform your ability to recruit after the pilot study is completed. Within the methods you should be more clear that a catheter will be placed. This again goes back to one of my concerns – you are potentially placing a catheter in people who may only have one fracture? Is it ethical to expose people to the risk of a catheter associated complication for 1 fracture? Page 11 line 3-4. “catheter programmed-intermittent boluses of 15ml 0.125% levo-bupivacaine given 3 hourly with option patient or clinician top-up of 5ml up to every 1 hour.” - Do you mean the bolus will occur 3 times an hour or every 3 hours? Clarify - Also, by “top-up” I assume you mean the patient can have an additional bolus of 5 ml every hour? Top-up seems like a rather unscientific term and perhaps a different word would be use.
---

	Other considerations. Have the patient administered control light illuminate as if they could request a bolus, just do not connect it to any fluid. Why not have a bolus of saline given at same amount as the study drug? Shouldn't another secondary outcome measure be successful placement of the catheter? How are you going to grade effectiveness of cough? This is a very subjective outcome... define it well. Are paracetamol or NSAID doses appropriate outcomes since they should be scheduled medications as opposed to "as needed" to be more effective. I am glad to know that this study is being done and look forward to the author's reply.
--	---

REVIEWER	Verma, Ruch Sanjay Gandhi Post Graduate Institute of Medical Sciences, Lucknow, Uttar Pradesh, India, Department of Anaesthesiology
REVIEW RETURNED	18-Jul-2022

GENERAL COMMENTS	Limitations of the study are not adequately discussed in Strengths and Limitations section. Please re-explain the blinding.
---

VERSION 1 – AUTHOR RESPONSE

Reviewer: 1

Dr. Thomas Carver, Medical College of Wisconsin

Comments to the Author:

I feel that the authors present a very clear description of their study and that it is an important topic that will add to the gap in literature. Recognizing that the study protocol is already in place I still have some suggestions/questions regarding the design that I feel should be reviewed.

Page 10 line 59. I do not know what a "guided ESP block story" is. Please clarify this sentence.

Thank you for highlighting this typographical error that we have corrected.

Study design. 1 rib fracture may be too inclusive. Additionally, what about location of fractures? Are you going to try to do a ESP on someone with a first rib fracture? Do you think the anesthetic will dissect up to that level?

I am worried that this will severely confound your study and I would suggest that 3 or more rib fractures be considered instead. You will be more likely to have a difference in clinical outcomes and this will also better inform your ability to recruit after the pilot study is completed.

We agree that including one rib fracture may be too inclusive. Equally, including a participant with 12 (or more) fractures may also be too inclusive. We consider it likely that ESP blockade, if it has any benefit to offer this patient group, is likely to be most advantageous to patients with 2-6 rib fractures. However this is speculative based on our local experiences, rather than based on robust evidence. A definitive trial will help answer this very question. An important qualifier is that the relationship between number of rib fractures and patient-experienced pain, is not straightforward or linear in our experience. Some patients with single rib fractures describe severe and disabling pain, while other patients with multiple fractures describe being relatively pain free. There are of course many reasons for this, not least pre-traumatic pain states and distracting injuries.

The competing need to design clinical trial methodologies to reflect broad and pragmatic inclusion criteria and, vitally, to facilitate timely trial recruitment, while ensuring that the intervention signal remains strong in the selected sample, is challenging. We felt a broad eligibility approach at feasibility and pilot phase was appropriate, so that we can use the pilot data to inform definitive trial design and justify any future restrictions on eligibility. Justification of restrictions on trial participation by eligibility

criteria is particularly important for patients, since any restriction inevitably denies some patients the opportunity to take part in research.

We acknowledge that a patient with an isolated first rib fracture could be included in the trial, provided they met other eligibility criteria. Apart from occasional first rib fractures co-existing with clavicular or scapula fractures, it is not a sub-group that we anticipate screening in large numbers for trial inclusion based on our local populations. There is no anatomical reason to suppose ESP spread would not be effective at the posterior rami at that level, and ESP blockade has been described, for example, for shoulder surgery.

Your comments on this matter are very valid and helpful and we will take these into consideration when we review the feasibility and pilot data from candidate clinical outcome measures when designing a definitive trial.

Within the methods you should be more clear that a catheter will be placed. This again goes back to one of my concerns – you are potentially placing a catheter in people who may only have one fracture? Is it ethical to expose people to the risk of a catheter associated complication for 1 fracture?

We have amended the wording to make it clearer that a catheter will be placed. The ethical approval for this study granted by NHS Research and Ethics did not raise a concern with catheters in patients with single rib fractures. Provided a participant has sufficient pain to meet eligibility criteria, then we consider a catheter technique reasonable. We believe pain should be the ultimate determinate of need, rather than numerical fractures. If a potential participant has only one rib fracture and this injury causes either mild or no pain, then they are ineligible for ESPEAR trial inclusion since clearly siting an ESP block (with or without catheter) is unethical in someone without sufficient pain to warrant an invasive procedure.

We consider the incidence of ESP-catheter associated complications to be low and the harm incurred usually minor. This is in contrast to epidural or paravertebral techniques, where catheter-associated harm is more common and carries a greater harm burden. The commonest ESP catheter complication (in our local experience) is leakage of local anaesthetic at the insertion site. The second is dislodgement. The feared complications of infection and migration we consider to be very low when inserted under US guidance under full aseptic technique. We will be collecting safety data in this pilot and in any future trial.

Page 11 line 3-4. “catheter programmed-intermittent boluses of 15ml 0.125% levo-bupivacaine given 3 hourly with option patient or clinician top-up of 5ml up to every 1 hour.”

- Do you mean the bolus will occur 3 times an hour or every 3 hours? Clarify
- Also, by “top-up” I assume you mean the patient can have an additional bolus of 5 ml every hour? Top-up seems like a rather unscientific term and perhaps a different word would be use.

We have clarified this wording and thank you for pointing it out.

Other considerations. Have the patient administered control light illuminate as if they could request a bolus, just do not connect it to any fluid. Why not have a bolus of saline given at same amount as the study drug?

Shouldn't another secondary outcome measure be successful placement of the catheter?

How are you going to grade effectiveness of cough? This is a very subjective outcome... define it well.

We considered a saline infusion delivered intra-fascially, as you describe. When considering the technique of blinding, we needed to balance trying to achieve effective blinding, while placing the minimum burden of potential harm to participants (and staff), and ensuring the trial would be feasible to deliver within reasonable resources. We considered that minimum harm would be achieved by not running saline infusions, but effectiveness of blinding would be maintained. As part of our embedded qualitative study we will interview participants and staff on blinding effectiveness. If participants tell us

our blinding attempt was unsuccessful, this will give us the justification at definitive trial to proceed to a more invasive blinding technique – for example the placement of a saline intrafascial catheter. If our blinding technique is effective then we will have avoided the need for a more invasive technique.

Are paracetamol or NSAID doses appropriate outcomes since they should be scheduled medications as opposed to “as needed” to be more effective.

We agree these will usually be prescribed by clinicians as regular medications, and we would not consider them outcomes as such, but they are collected in the trial since this helps contextualise opioid and ketamine usage. Their use is not protocolised in the study. We felt that completeness required us to capture this data, particularly for NSAIDs, since their use is variable in UK.

I am glad to know that this study is being done and look forward to the author's reply.

We would like to thank you for taking time to review the protocol and will consider all of your points going forward, even if some cannot be changed in the protocol at this stage.

Reviewer: 2

Dr. Ruch Verma, Sanjay Gandhi Post Graduate Institute of Medical Sciences, Lucknow, Uttar Pradesh, India

Comments to the Author:

Limitations of the study are not adequately discussed in Strengths and Limitations section. Please re-explain the blinding.

Thank you for these comments. We have amended the ‘Strengths and Limitations’ section and the ‘Blinding’ section to explain further the blinding technique. We have also listed a further important study limitation in the ‘Strengths and Limitations’ section as requested.

VERSION 2 – REVIEW

REVIEWER	Carver, Thomas Medical College of Wisconsin, Surgery
REVIEW RETURNED	27-Aug-2022
GENERAL COMMENTS	Thank you for addressing my concerns/comments. I have no additional comments.